# Towards Understanding Why Group Robustness Methods Work

## Abstract

Deep Learning has made remarkable strides, yet models trained under conventional Empirical Risk Minimization (ERM) approaches encounter challenges regarding their generalization capabilities. In particular, a lack of robustness to spurious correlations. In response, Group Robustness Methods (GRMs) have been developed to combat them. These methods partition training datasets into distinct groups based on spurious features and class labels and adjust their weighting in the loss function. These methods show remarkable performance in dealing with spurious correlations. The underlying mechanisms for their success, however, are not so well understood. Our work contributes by shedding light on the learning dynamics of GRMs, through an empirical and theoretical analysis of them that reveals the differences in feature learning and the type of classifiers they learn versus ERM. Surprisingly, both GRMs and ERM models retain spurious information in their representations, even when it is irrelevant to the task at hand. We find evidence that suggests that the key to GRMs' success is two-fold: distributing prediction across multiple features in representation space to avoid relying on few but spurious attributes and incentivizing the classifier to become orthogonal to spurious features. We verify our findings by proposing an upgrade to the Subsampling baseline method called Group Distributionally Robust Feature Reweighting (GDRFR) that is easy to compute and only requires a fraction of group labels during a finetuning phase and retrieve most of GRMs performance gains over ERM.

## 1 Introduction

In the past decade (Bengio et al., 2014), Deep Learning has witnessed remarkable progress. However, models trained using the conventional Empirical Risk Minimization (ERM) (Vapnik, 1991) approach encounter various challenges. Specifically, by prioritizing the reduction of training error, these models often learn features optimized for this objective, sacrificing their generalization capabilities in underrepresented groups. An exemplar of this phenomenon is the case of a model trained to distinguish between cows and camels. When confronted with test images where the cows are on a beach, and camels in grass, the model fails utterly. This is because the model has learned to use the simpler background features, instead of the shape of the animal to make its classification. This issue presents itself in the literature under different names such as spurious correlations (Arjovsky et al., 2020), simplicity bias (Shah et al., 2020), or shortcut learning (Geirhos et al., 2020). It is not related only to the image domain but to other modalities as well (Williams et al., 2018; Pavlopoulos et al., 2020).

To tackle this problem, some approaches have been proposed for enhancing model robustness. Among these, Group Robustness Methods (GRMs) have demonstrated state of the art performance. Typically, these methods involve partitioning the training dataset into distinct groups, often delineated by the presence of spurious features and class labels and optimize to maximize the Worst Group performance Hu et al. (2018). Group Distributionally Robust Optimization (GDRO) (Sagawa* et al., 2020) is one of the most effective among them, with many subsequent works following in its steps to try and overcome its drawbacks (Seo et al., 2022; Sohoni et al., 2020; Liu et al., 2021; Idrissi et al., 2022; Kirichenko et al., 2023).

Yet, despite these recent advances, there remains a *limited understanding of the inner workings* behind current state-of-the-art models. In this context, our work contributes by shedding light on the learning

dynamics of GRMs. We study what makes them so successful through an empirical and theoretical lens. Our study is conducted in a controlled and limited setting, that is of vision datasets, but we believe that our findings provide valuable insights that can motivate further research on how current machine learning models handle relevant problems, such as learning in the presence of spurious correlations. We believe that more research focused on understanding these mechanisms is crucial for the advancement of our field.

Our study begins by understanding what is learned in representations trained through GRMs. Our first hypothesis is that spurious features are ignored from their representations. Without spurious information in the representation, the model would necessarily be robust to spuriousness. Surprisingly, we find that both GRMs and ERM still learn spurious features, *even when they are not relevant for the task at hand*. We then turn to the role of the final linear classifier of a model. We show both theoretically and empirically, that GRMs produce classifiers that will tend to be more orthogonal to directions in representation space aligned with spurious attributes than classifiers produced by ERM. However, we find this effect on its own is not enough to explain GRMs' success.

Finally, we test the following hypothesis: GRMs work by disentangling spurious attributes from invariant ones in representation space, allowing the classifier to ignore said spurious attributes by becoming orthogonal to them. Surprisingly, we find the opposite is true: ERM produces more disentangled features, however, GRMs tend to rely on multiple features for prediction, making it more robust to rely on spurious ones, while ERM relies on less.

We also study Subsampling (Idrissi et al., 2022), a method that achieves good results without any feature learning whatsoever. This method has allowed for the development of more practical GRMs, in particular, Deep Feature Reweighting (Kirichenko et al., 2023). We find evidence that this method is mostly supplementary to the feature-learning effects of GRMs. These findings allow us to propose a new competitive and simple baseline: Group Distributionally Robust Feature Reweighting: a method that only requires group labels during a single finetuning pass, with reduced data requirements as well. It retrieves most of GDRO's performance gains over ERM (at least 92% of them) and sometimes even surpasses it.

Thus, our contributions can be summarized as follows:

- We show that ERM and GRM models tend to learn spurious information in their representations in spite of them not being useful for the current task or being trained with specific methods to mitigate performance based on such features.

- Group Robustness Methods such as Group DRO or Reweighting work by aligning the classifier so it ignores spurious information in the learned representation. We show empirical evidence for this and show mathematical results that suggest why this should be so for the spurious correlation setting. Our theoretical results hold for an arbitrary non-linear SGD trained model.

- We find evidence that suggests the difference between ERM and GRMs features lies in that GRMs distribute predictive features across many dimensions while ERM tends to produce representations that rely on less features for prediction. Surprisingly, Disentanglement is higher for ERM than GRMs.

- We find evidence that suggests that Subsampling partially retrieves GRMs feature learning capabilities, or put in other words, is supplementary to the feature learning effects of GRMs. GRMs seem to learn specific features that ERM does not capture.

- Our findings derive in an improved baseline that retrieves most of GDRO's performance gains while only requiring a fraction of group labels *for finetuning* which requires only a single finetuning pass.

## 2 PRELIMINARIES

Our aim is to determine the origins of the benefits of GRMs. To begin this task, we will first specify our problem setting.

## 2.1 PROBLEM FORMULATION

We'll work on a binary classification problem with input features $\vec{x} = \{\vec{x}_{inv}, \vec{x}_{sp}\}$, where $\vec{x}_{inv}$ represents invariant features and $\vec{x}_{sp}$ is an spurious attribute correlated with correlation $c$ with the ground truth label $y$. We will partition our dataset into $G = 4$ groups with respect to $\vec{x}_{sp}$ and $y$. Let $\mathcal{G} = \{s0, s1, n0, n1\}$. These are groups defined by whether the spurious correlation is predictive ("s") of the ground truth label or not ("n"), and the value of the ground truth label (0,1). Let $N_g$ be the size of group $g$ and $N = \sum_{g \in \mathcal{G}} N_g$ is the total amount of samples.

We will also assume that we will train a model on this problem using SGD, and that this model can be separated into an arbitrary non-linear feature extractor $\Phi$ and a linear classifier $C(\vec{x}) = W \cdot \vec{x} + \vec{b}$. The output of the feature extractor will be a set of features that depend on the invariant and spurious features of the data = $\Phi(\vec{x}_{inv}, \vec{x}_{sp})$. We will assume the spurious feature may take two values: $\{\mathcal{B}, -\mathcal{B}\}$. Our final model then takes the form $M(\vec{x}) = C(\Phi(\vec{x}))$

Spurious Vector. Let $\lambda = \vec{v} \cdot \vec{x}$. Define vector $\vec{v}$ as *spurious w.r.t* $\mathcal{G}$ if:

$$\vec{x} \in \{s0, n1\} \implies \lambda > 0, \vec{x} \in \{s1, n0\} \implies \lambda < 0$$

This definition intuitively indicates that *$\vec{v}$ is a useful direction for classification in groups where the spurious correlation holds*, and is not useful where it doesn't hold.

## 2.2 DATASETS

**MNIST-CIFAR (Shah et al., 2020)**    MNIST-CIFAR consists of images from MNIST (LeCun & Cortes, 2010) and CIFAR-10 (Krizhevsky et al.) concatenated vertically while a spurious correlation is induced by associating CIFAR-10 labels to MNIST labels with a tunable correlation parameter. Classes 0 and 1 from MNIST are correlated with corresponding classes of CIFAR-10. We train our models on versions of this dataset with correlations of 0.0, 0.25, 0.5, 0.75 and 0.9.

**Waterbirds (Sagawa* et al., 2020)**    Waterbirds is a dataset of real images of birds where a spurious correlation is induced with the background. The task consists of identifying if the bird is a land or sea bird, while the background is land or sea. This dataset shows 0.9 correlation between class label and spurious attribute, so we use their code to generate versions for correlations of 0.0, 0.25, 0.5, 0.75 and 0.9.

**CelebA (Liu et al., 2015)**    CelebA is a dataset of real images of celebrities carefully annotated with different attributes (gender, hair color, facial hair, attractiveness, etc.) which allows for extensive creation of spuriously correlated datasets. We use the same splits and attributes as in Sagawa* et al. (2020), where the target attribute is Blonde Hair and the spurious attribute is gender. We only use the original training split for this dataset, which is 0.3. However, for ease of display on tables and figures, we list all results for CelebA under correlation = 0.9.

## 2.3 MODELS

For MNIST-CIFAR we use a simple convolutional network with 3 convolutional layers of 32, 64, 128 filters and a linear layer at the end, with ReLU activations. No Max-Pooling was used. For Waterbirds and CelebA we use a Resnet-50 (He et al., 2016) pretrained on ImageNet (Deng et al., 2009).

## 2.4 ROBUSTNESS METHODS

Group Robustness Methods (GRMs) are based around the idea of reweighting examples from the training set. The most successful methods like Group DRO and Reweighting assume the training distribution to be a mixture of $G$ groups. In practice, what they do is to create $G = |\mathcal{A}| \times |\mathcal{Y}|$ groups. Where $|\mathcal{A}|$ is the number of possible values an input spurious attribute might have, while $|\mathcal{Y}|$ is the number of classes in the classification problem. To define these groups requires previous knowledge of spurious correlations in the data. Plenty of methods attempt to forego this requirement for group labelling in the training phase by finding proxies for it through clustering (Seo et al., 2022) or other means Liu et al. (2021).

The most general form of these methods has the following Loss Function:

$$\mathcal{L}^{GEN} = \sum_{i=1}^{N} p_i \cdot \mathcal{L}(x_i, y_i)$$

Where $p_i$ is simply a weighting factor for the $i-$th example which can be either a constant or more complex function. Through the rest of this article, we will work with 4 methods: ERM, Reweighting, Group DRO and Subsampling.

### 2.4.1 EMPIRICAL RISK MINIMIZATION (ERM)

This is the traditional training loss where the average loss across the training data is minimized. This method suffers heavily from spurious correlations. Its loss function becomes:

$$\mathcal{L}^{ERM} = \sum_{i=1}^{N} \frac{\mathcal{L}(x_i, y_i)}{N}$$

### 2.4.2 REWEIGHTING

This GRM (Shimodaira, 2000) involves reweighting the loss of each group in the training data, so as to mitigate the impact of each group's size. In particular, the loss function becomes simply the average loss of each group's average loss. $G$ is the number of groups in the dataset.

$$\mathcal{L}^{RW} = \frac{1}{G} \sum_{g \in \mathcal{G}} \sum_{x_i \in \mathbb{X}_g} \frac{\mathcal{L}(x_i, y_i)}{N_g}$$

### 2.4.3 GROUP DRO

Group Distributionally Robust Optimization (GDRO) (Sagawa* et al., 2020) tackles worst-group error by optimizing the following function ($\epsilon$ is a hyperparameter) during training:

$$\mathcal{L}^{GDRO} = \sum_{g \in \mathcal{G}} p_g \sum_{x_i \in \mathbb{X}_g} \frac{\mathcal{L}(x_i, y_i)}{N_g}, p_i = \frac{e^{\epsilon \cdot \mathcal{L}_i}}{\sum_{i=1}^{G} e^{\epsilon \cdot \mathcal{L}_i}}$$

### 2.4.4 SUBSAMPLING (SUBG-FT)

An alternate and simple baseline for inducing robustness is by using Subsampling (Idrissi et al., 2022), which consists of training a model using ERM on a subsampled version of the dataset which enforces balance between groups. Usually, each group is subsampled to the size of the smallest group.

Our implementation of these methods is based off of (Liu et al., 2021). With these preliminaries addressed, we proceed to our initial inquiry into the mechanisms by which GRMs operate.

Finally, we do not include other methods such as JTT (Liu et al., 2021), DFR (Kirichenko et al., 2023), AFR (Qiu et al., 2023) or LFR(Ghaznavi et al., 2023) in our analysis because these methods in one way or another are a combination of the simpler methods studied here; they estimate weights for each sample either through proxies for group labels and/or they use Subsampling for finetuning a classifier. Our interest lies in understanding mechanisms rather than a complete review of available methods.

## 3 DO GRMS ELIMINATE INFORMATION ABOUT SPURIOUS FEATURES?

We will analyze first if the representations learned by GRMs contain information about spurious features. One hypothesis about their success might relate to them discarding this information during training. To do this, we obtain the singular vectors of a PCA decomposition of representations of the training set. For each singular vector we train a logistic regression to predict the spurious label on the training set and then evaluate on the test split. Figure 1 shows these results for all datasets. For models trained on MNIST-CIFAR, the first direction is surprisingly predictive of the MNIST label both for the training set and the test set, independently of which method was used to train. What is

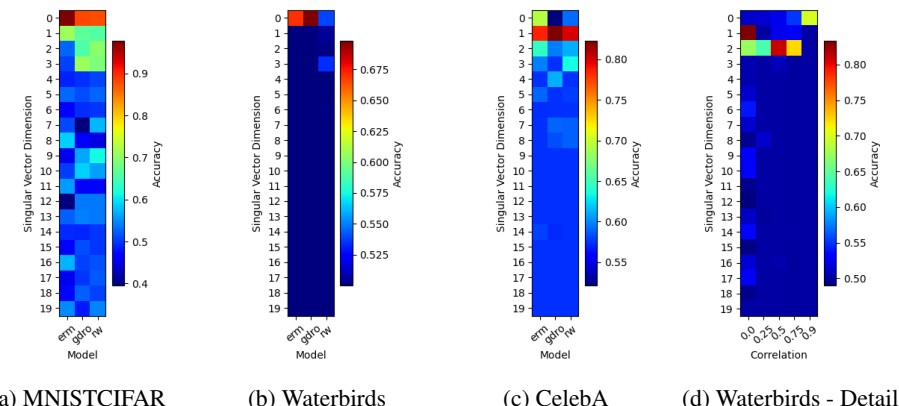

(a) MNISTCIFAR      (b) Waterbirds      (c) CelebA      (d) Waterbirds - Details

Figure 1: (a-c) Test accuracy for predicting the spurious label (corr=0.9) for the top 20 Singular Vectors obtained from representations derived by training using different methods. Representations from models trained in MNIST-CIFAR, Waterbirds and CelebA. For all methods, we find directions that are surprisingly predictive of the spurious label. Moreover, these directions are within the top-3 singular vectors, suggesting their relevance. (d) We also observe that even when the spurious correlation is not useful (corr=0), it is still learned in the representation for MNIST-CIFAR and Waterbirds.

remarkable is that this happens even if the spurious label is not actually useful for the task: models trained on datasets with no correlation between the spurious and target label still retain information about the spurious label. This behaviour happens also in more complex datasets like Waterbirds and CelebA, with the first and second directions, respectively being highly predictive of the spurious label. This behavior is consistent across all methods, which suggests that all of them store spurious information in their representations and there is not much of a mechanism to curb this.

Therefore, it may just be that the quality of features and not their spurious content is the reason for GRMs' success. We test this by training models on GDRO and finetuning a new classifier using ERM. Results are shown in Table 2 (ERM-FT). As can be seen, performance is usually worse than ERM. These experiments suggest that representations learned by all methods store spurious information and are not themselves enough to explain their success. We proceed with our analysis by studying the impact of these methods on the final classifier layer $C$.

## 4   WHAT IS LEARNED IN THE FINAL CLASSIFIER LAYER BY EACH METHOD?

Since spurious information is still retained in the representation, independently of the method used for training, we must suppose that the *classifier learns to ignore this information when training under a GRM*. How can a linear classifier do this? A straightforward explanation is that the classifier becomes more orthogonal to the spurious direction $\vec{v}$. Therefore we would like to understand how the dot product between $\vec{v}$ and the classifier learned by each method changes during training. This quantity gives us a measure of how relevant the spurious direction is for the classifier.

### 4.1   THEORETICAL STUDY ON CLASSIFIER ALIGNMENT

Let $\mathcal{L}(x_i, y_i)$ be our loss function, which we will assume to be Cross Entropy, the most widely used loss function for classification problems. Let $\mathbb{X}_g = \{\Phi(x)\colon x \in g\}$ be the set of representations of samples from group $g$. For simplicity, let's assume $C$ has no bias term (i.e. $C(\vec{x}) = W \cdot \vec{x}$). Then, let $\vec{v}_{sp}$ be a spurious vector respective of $\mathcal{G}$ and $\alpha_{sp} = \vec{v}_{sp} \cdot W$ be the *alignment* of $\vec{v}_{sp}$ respective of the linear classifier $W$. For each training iteration $t$ of SGD with learning rate $\mu$, $\alpha_{sp}$ changes by the following amount:

$$\frac{\Delta \alpha_{sp}}{\Delta t} = \vec{v}_{sp} \cdot (W^{t+1} - W^t) = -\vec{v}_{sp} \cdot \mu \cdot \frac{\partial \mathcal{L}}{\partial W^t}$$

Therefore, $\frac{\Delta \alpha_{sp}}{\Delta t}$ depends only on $\frac{\partial \mathcal{L}}{\partial W^t}$ for each method and we will proceed to study these terms.

Let $\mathcal{L}_g = \sum\limits_{x_i \in \mathbb{X}_g} \frac{\mathcal{L}(x_i, y_i)}{N_g}$ be the mean loss for group $g$. Through a simple derivation (see section F.1 in the Appendix) we obtain the following expressions for:

$$\frac{\partial \mathcal{L}^{ERM}}{\partial W} = \sum_{i=1}^{N} \frac{(s_i - y_i) \cdot x_i}{N},$$

$$\frac{\partial \mathcal{L}^{RW}}{\partial W} = \frac{1}{G} \sum_{g \in \mathcal{G}} \bar{x}_g \quad \text{and}$$

$$\frac{\partial \mathcal{L}^{GDRO}}{\partial W} = \sum_{g \in \mathcal{G}} \bar{x}_g \cdot (l_g + p_g).$$

Where $x_j$ is the j-th element of $\mathbb{X}_g$, $s_j$ is the softmax output of $x_j$; $\bar{x}_g = \sum\limits_{x_i \in \mathbb{X}_g} \frac{(s_i - y_i) \cdot x_i}{N_g}$ is a weighted average of the $x_j$ for group $g$; $l_g = \epsilon \sum\limits_{j \in \mathcal{G}} p_j \cdot (\mathbb{1}_{g=j} - p_g)\mathcal{L}_j$ is a weight associated to group $g$ by its loss.

With all of this groundwork done, we can state our first theoretical result:

**Theorem 4.1.** *Define*

$$\frac{\Delta \alpha_{sp}}{\Delta t}^{ERM} := \sum_{g \in \mathcal{G}} \sum_{j=1}^{|\mathbb{X}_g|} \frac{\lambda_j \cdot (s_j - y_j)}{N}$$

$$\frac{\Delta \alpha_{sp}}{\Delta t}^{RW} := \sum_{g \in \mathcal{G}} \sum_{j=1}^{|\mathbb{X}_g|} \frac{\lambda_j \cdot (s_j - y_j)}{4 N_g},$$

*where $\lambda_j = \vec{v}_{sp} \cdot x_j$, $N_g$ is the size of group $g$ and $N$ is the total amount of samples. Then, $\frac{\Delta \alpha_{sp}}{\Delta t}^{ERM} > \frac{\Delta \alpha_{sp}}{\Delta t}^{RW}$.*

This result shows that classifiers trained on Reweighting align themselves to an spurious direction $\vec{v}_{sp}$ less than a classifier trained on ERM on each training iteration. Crucially, this only depends on the size of the groups, specifically that groups where the spurious correlation is useful are over-represented in the training data.

**Remark 4.1** (Case of GDRO.). *This result does not hold for all cases when using Group DRO. However, we can show that for our scenario it will usually hold. The previous result holds as long as the following conditions hold:*

- *Case 1:*

$$\frac{N_g}{N} > l_g + p_g, \quad g \in \{G_{s0}, G_{s1}\}$$

  *In this case $\frac{N_g}{N} > p_g$ will usually hold as the group will be overrepresented in the dataset, which should make its average loss be below the total average loss. Therefore, this condition will mostly hold when $l_g < 0$. This happens for group $i$ when:*

$$\left(\sum_{i \neq j} p_j\right) \cdot \mathcal{L}_i < \sum_{i \neq j} p_j \cdot \mathcal{L}_j.$$

  *Which in conventional language means that if the i-th group's average loss is below the average of average losses of the rest of the groups, $\frac{\Delta \alpha_{sp}}{\Delta t}^{ERM} > \frac{\Delta \alpha_{sp}}{\Delta t}^{GDRO}$ will hold.*

- *Case 2:*

$$\frac{N_g}{N} < l_g + p_g, \quad g \in \{G_{n0}, G_{n1}\}$$

  *Through a similar argument, $\frac{N_g}{N} > p_g$ as $g$ will be underrepresented in the dataset and $l_g$ will have to be positive, which it will usually be as the average loss will tend to be greater for underrepresented groups. Thus, $\frac{\Delta \alpha_{sp}}{\Delta t}^{ERM} > \frac{\Delta \alpha_{sp}}{\Delta t}^{GDRO}$ also holds in this case.*

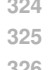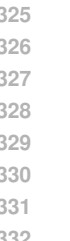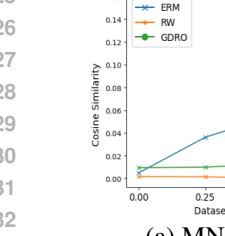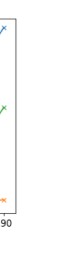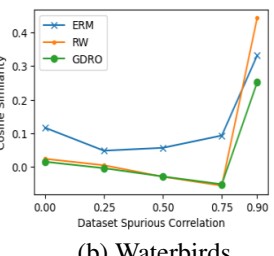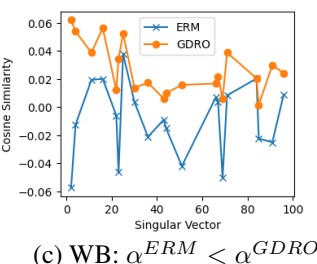

(a) MNIST-CIFAR         (b) Waterbirds         (c) WB: $\alpha^{ERM} < \alpha^{GDRO}$

Figure 2: (a-b) Cosine Similarity between the classifiers of models trained on different methods versus the most spurious singular vector in the representation. Our hypothesis is that ERM classifiers will align themselves more to spurious directions than GRM methods ($\alpha^{ERM} > \alpha^{GDRO}$). We observe that it is the case for both MNIST-CIFAR and Waterbirds. This holds even for different levels of spurious correlations in the training data. (c) As a sanity check, we review the cosine similarities for other singular vectors where this relationship doesn't hold.

In summary, RW balances losses only using group sizes, while GDRO offers an additional mechanism that takes into account each group's average loss, so it can even work in cases where group sizes are balanced, but their losses are not. In the following section, we proceed to evaluate if there's empirical evidence for these theoretical results.

## 4.2 EMPIRICAL EVALUATION OF CLASSIFIER ALIGNMENT

First, we empirically check the alignment of the resulting classifiers for ERM, RW and GDRO. Results can be seen in Figure 2. We observe that while all methods show an increase of alignment as we increase the spuriousness of the dataset, the alignment of ERM is almost always greater than that of RW and GDRO. This suggests positive evidence for our hypothesis that GRMs act on the classifier to allow it to ignore spurious features.

## 4.3 EMPIRICAL EFFECTS OF FINETUNING FINAL CLASSIFIER LAYER ON GRMS

A necessary path forward is to test how important this classifier effect is. If most of the task of achieving robustness is done by the final linear classifier, then it could allow us to forego the need for feature learning using group labels. To test this, we train models on ERM then train the classification heads using ERM, RW and GDRO for all datasets. Results are shown in Table 1. For MNIST-CIFAR and CelebA, we can retrieve most of the gains from using GDRO, sometimes even surpass them, simply by retraining the classifier using GDRO based on ERM features. This effect, however, dissipates when testing on Waterbirds, a more complex dataset where using ERM features with GDRO on the classifier yields similar results to ERM. If the spurious information is retained for all methods, but the classifier alone cannot ignore it, then it can mean at least one of two things: that spurious features are more entangled with the invariant features when training with ERM and/or that even if disentangled, ERM has no incentive to use the invariant features. Supporting evidence for the latter assumption is given by previous experiments as ERM-FT performs equally or worse than ERM when finetuned using ERM. Therefore, to understand our failure in Waterbirds, we come back again to studying representations, in particular, how different methods affect disentanglement.

## 4.4 TESTING FOR DISENTANGLEMENT OF FEATURES

We proceed to quantitatively study disentanglement directly. To do so, we leverage previous work on metrics for disentanglement (Eastwood & Williams, 2018). In particular, we use the framework of DCI metrics: Disentanglement, Completeness and Informativeness. *Disentanglement* is a score that quantifies how many latent attributes (spurious and class labels, in our case) a particular dimension is predictive of. A score of 0 indicates that each dimension is equally predictive of all latent attributes, while a score of 1 indicates that each dimension is predictive of solely one latent attribute. *Completeness* is a score that quantifies how many dimensions from the representation are predictive of a particular latent attribute. A score of 0 indicates that each latent attribute is equally predicted by all

Table 1: Worst Group Accuracy results for various methods trained on both MNIST-CIFAR and Waterbirds benchmarks on different levels of spurious correlation between class label and spurious attribute and CelebA for a correlation of 0.9. Results correspond to evaluation on a test dataset with no correlations between class and the spurious attribute. Three seeds were used. Finetuning the classifier with GRMs over ERM features retains most of the effect of the GRMs on MNIST-CIFAR and CelebA, while it works poorly in Waterbirds. This suggests that while it is an important effect, representations learned by GRMs are also important.

| DATASET | METHOD | CORRELATION | | | | |
|---|---|---|---|---|---|---|
| | | **0.0** | **0.25** | **0.5** | **0.75** | **0.9** |
| MNIST-CIFAR | ERM | 88.62% (0.93) | 82.38% (0.62) | 78.21% (0.89) | 55.23% (5.18) | 30.17% (2.99) |
| | RW | **89.16%** (0.70) | **88.62%** (1.08) | 89.51% (0.93) | **86.35%** (0.98) | **85.87%** (0.85) |
| | GDRO | 88.09% (1.23) | 88.21% (1.08) | 88.99% (0.95) | 85.49% (0.93) | 85.24% (1.14) |
| | RW-FT | **89.16%** (1.20) | 88.08% (1.02) | **89.74%** (0.22) | 85.42% (0.26) | 82.61% (0.56) |
| | GDRO-FT | 88.76% (1.06) | 87.94% (1.02) | 89.38% (0.77) | 86.10% (0.93) | 82.74% (0.99) |
| | SUBG-FT | - | 88.76% (0.00) | 87.82% (0.46) | 86.48% (0.46) | 82.48% (1.28) |
| | GDRFR (RW) | - | 89.02% (0.23) | 88.62% (0.23) | 86.61% (0.23) | 82.46% (1.62) |
| | GDRFR (GDRO) | - | 88.76% (0.40) | 88.22% (0.46) | 86.61% (0.61) | 82.74% (1.06) |
| WATERBIRDS | ERM | 89.36% (0.55) | 89.72% (0.41) | 87.95% (0.63) | 81.62% (0.87) | 72.95% (0.36) |
| | RW | 89.62% (0.24) | 90.91% (0.50) | 89.77% (0.63) | 88.06% (1.64) | 86.31% (0.30) |
| | GDRO | **90.86%** (0.18) | 91.69% (0.09) | 90.45% (0.09) | 89.30% (0.09) | 87.31% (0.27) |
| | RW-FT | 84.32% (0.78) | 86.55% (0.50) | 82.71% (0.16) | 75.34% (1.15) | 66.20% (2.16) |
| | GDRO-FT | 86.71% (0.50) | 87.28% (0.18) | 85.20% (0.56) | 79.08% (1.60) | 73.78% (2.92) |
| | SUBG-FT | - | 90.76% (0.09) | 90.13% (0.55) | 89.56% (0.78) | 87.69% (1.09) |
| | GDRFR (RW) | - | 91.07% (0.24) | 91.12% (0.47) | **90.91%** (0.80) | 88.89% (0.78) |
| | GDRFR (GDRO) | - | **91.43%** (0.41) | **91.48%** (0.59) | 90.86% (0.74) | **88.99%** (0.65) |
| CELEBA | ERM | - | - | - | - | 48.15% (2.10) |
| | RW | - | - | - | - | 89.93% (0.72) |
| | GDRO | - | - | - | - | **91.05%** (0.47) |
| | SUBG-FT | - | - | - | - | 87.22% (0.56) |
| | RW-FT | - | - | - | - | 88.61% (0.74) |
| | GDRO-FT | - | - | - | - | **91.06%** (0.47) |
| | GDRFR (RW) | - | - | - | - | 86.48% (1.40) |
| | GDRFR (GDRO) | - | - | - | - | 90.10% (0.42) |

dimensions of the representation while a score of 1 indicates solely one dimension predicts each latent attribute. *Informativeness* just quantifies how predictive the representation is of the latent attributes. This is what has been traditionally reported when analyzing representations through training of linear probes. Theoretically, a good representation would have high disentanglement, high completeness and high informativeness. However, results challenge this notion in the case of spurious correlations. We perform experiments for all different methods across all datasets. Figure 3 shows results for MNIST-CIFAR and Waterbirds. Disentanglement decreases as spurious correlation increases which is to be expected. Results for disentanglement across methods, however, are unexpected. Disentanglement is actually consistently higher for ERM than for GRM methods. Completeness, however, is the key difference: *Completeness is consistently lower* across all datasets for GRMs than for ERM. This implies that representations derived from GRMs are more robust because they allow predictability of attributes to rely on more dimensions, which in the case of spurious correlations is highly relevant, as one (spurious) dimension is too predictive. See Table 4 in the Appendix for expanded results, including CelebA. Thus, what is expected of a good representation in the general case, may not necessarily be desirable in the spurious correlation scenario, where having multiple dimensions that are predictive of a latent attribute (i.e. low completeness) aids in making the model more robust to focusing on potentially spurious dimensions for prediction.

### 4.5 SKIPPING GRMS FOR REPRESENTATION LEARNING

Interestingly, as we discussed in Section 2.4.4, there exists a baseline method which has been applied in recent work (Kirichenko et al., 2023) with great success, which finetunes a classifier using SUBG,

along with feature normalization, L1 loss and an ensemble method akin to model soups (Wortsman et al., 2022). This suggests a path forward to dispense with group labels during training of the full model. Therefore, it would seem to pose an alternative to the representation-learning effects of GRMs. As our aim is to understand better how these methods work, we use a simplified version of this method that only uses SUBG to finetune ERM features. We dub this method SUBG-FT.

Table 1 shows that SUBG-FT is a massive improvement over ERM on all datasets, but underperforms against GDRO, except in Waterbirds. We believe this is because of GDRO's effect on the classifier which is better at ignoring spurious attributes. Therefore, finetuning a classifier on features derived from ERM combining SUBG-FT and GDRO should yield most of GDRO's performance, which is our proposed new baseline method: Group Distributionally Robust Feature Reweighting. This is listed as "GDRFR" in Table 2. We observe that indeed a performance of $88.99\%$ is achieved on Waterbirds (c=0.9) just finetuning the classifier, 2% over performance from full training based on GDRO. This is relevant, as we used around 5% of the training data with group labels which are both necessary for SUBG-FT and GDRO to finetune the classifier, allowing for practical uses on already deployed systems. For MNIST-CIFAR and CelebA, GDRFR retrieves most of the performance of GDRO, (94,38% and 97,79% respectively for the hardest setting). In Waterbirds and CelebA, we see that GDRFR significantly outperforms SUBG-FT obtaining around 10% and 6% more gains respectively. Please see Table 5 in the Appendix for detailed information. An important practical observation, is that for these methods to perform well, the *finetuning data should be different than the training data*. Finetuning on the same training data does not seem to work well.

Finally, we perform ablations on the size of the finetuning data. Results are shown in Table 3 in the Appendix. GDRFR which already uses little data (around 5-10% of the original training data size), works reasonably well with even 50% of that data, suggesting that it may work well in practical applications.

### 4.5.1 SUBSAMPLING IS SUPPLEMENTARY TO GDRO'S FEATURE LEARNING EFFECTS

Perhaps using SUBG-FT is just a complementary technique to GDRO. To test this, we train a classifier using GDRFR (GDRO) on GDRO features. Table 2 shows that this (GDRO + GDRFR(GDRO)) does not always work better than GDRO. In fact, when it does, the standard deviations are too high to assert this with statistical significance. This suggests that whatever gains are made by using Subsampling in the finetuning stage are achievable from better representations. That is to say, Subsampling seems to be supplementary to GDRO's feature learning effects.

## 5 RELATED WORK

### 5.1 METHODS FOR ROBUST LEARNING

Invariant Risk Minimization (Arjovsky et al., 2020) modifies the loss function to make the model invariant to different environments. Group Reweighting schemes manipulate the importance of samples during training: Reweighting (Shimodaira, 2000) reweights group to eliminate the impact of group size on the loss, Group Distributionally Robust Optimization (GDRO) (Sagawa* et al., 2020) reweights group losses based on their magnitude, while other methods (Seo et al., 2022; Sohoni et al., 2020) try to apply GDRO by finding proxies for the group labels needed. Other methods are based on multiple training passes: some finetune an ERM trained model on non correlated data (Kirichenko et al., 2023; Qiu et al., 2023; Ghaznavi et al., 2023) or use that same model to find samples to upweight to train a new model from scratch (Liu et al., 2021) or finetune a classifier. Others have worked on creating non-linear classifiers that are orthogonal to a set of attributes (Xu et al., 2022), this requires having a notion of both the train and test distribution of those attributes. Other methods have used contrastive losses (Zhang et al., 2022) to create representations that are invariant to spurious attributes. Finally, some have used Self Supervised pretrained models to estimate a logit adjustment term on the loss (Tsirigotis et al., 2023).

### 5.2 STUDIES ON ROBUSTNESS METHODS

Shah et al. (2020) show that neural networks are extremely susceptible to simplicity bias, even if core features are more predictive than simpler spurious features. Other work (Kirichenko et al., 2023)

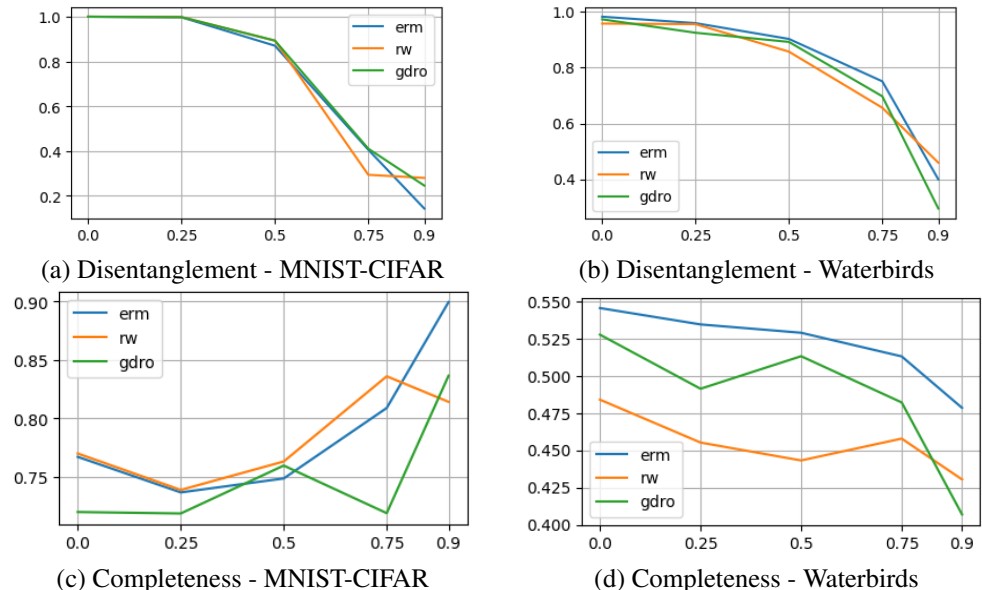

Figure 3: Disentanglement and Completeness metrics for representations derived from ERM, Reweighting and GroupDRO for various levels of spurious correlation between spurious attribute and class label. ERM consistently shows both higher Disentanglement (a-b) and Completeness (c-d) than the GRMs. Lower Completeness for GRMs suggests that GRMs distribute more predictive features across different dimensions in the representation, making it rely less on spurious features solely.

perform studies on representations derived from ERM-trained models, while we study representations learned by models trained with robustness methods.

Others (Ahmed et al., 2021) explore performance of different methods (among them GDRO) with a focus on systematic generalization and semantic anomaly scenarios, but do not report findings on the mechanisms by which that performance is achieved, unlike our work. Other work Zhang et al. (2022) draws connections between the relative mutual information of representations between spurious attributes and class labels with worst group error. They also propose alignment as a factor to explain the success of robustness methods, but their notion of alignment is between samples from the same class but different spurious attribute in representation space. Instead, we measure alignment between a model's classifier and a particular (spurious) direction in representation space.

## 6 CONCLUSIONS

In this work, we present a study on the mechanisms by which Group Robustness Methods are successful. We arrive at a few conclusions: they learn representations that, while surprisingly retaining spurious information, seem to distribute predictive features along many dimensions to achieve robustness to using just few spurious features; second, their loss function induces a behaviour in the classifier to learn to ignore spurious attributes more than ERM; this effect is compounded by the representation learning aspect of these methods. This is shown both empirically and theoretically. We also do a comparison against a Subsampling baseline which seems to achieve good results without any representation learning whatsoever. We find evidence that the effects of Subsampling seem to be supplementary to those representation learning effects of GRMs. Finally, through all of these findings we propose a simple new baseline that retrieves most of GDRO's performance gains that requires group labels only during a finetuning phase and using less than 5% of the original training data.

## 7 REPRODUCIBILITY STATEMENT

We share our code for our experiments, which is modified from Liu et al. (2021), in the supplementary materials. Proof for results on the theoretical section are found in the Appendix, Section F.1.

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

## A  DATASET EXAMPLES

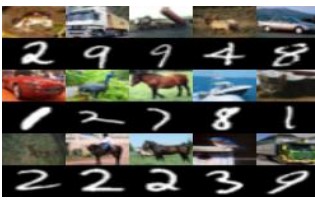
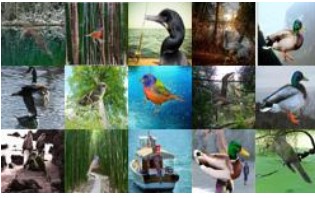
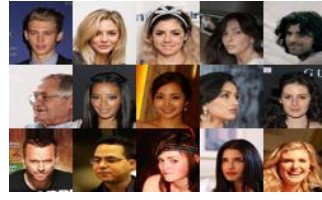

| (a) MNIST-CIFAR | (b) Waterbirds | (c) CelebA |
|---|---|---|

Figure 4: Sample images from datasets used in this work. MNIST-CIFAR correlates MNIST digits with CIFAR-10 classes; Waterbirds correlates land/water birds with a land/water background; CelebA consists of heavily annotated images from celebrities. In this work, we use the correlation that appears between gender and hair color.

## B  ADDITIONAL RESULTS

Table 2: Worst Group Accuracy results for various Resnet50 models trained using different methods on the Waterbirds benchmark on different levels of spurious correlation between class label and spurious attribute on the test set. We compare different finetuning strategies using different starting features and different loss functions. All results are reported using 3 seeds. Training methods are classified according to the following criteria: **FRZX**: convolutional layers from X onwards are unfrozen. **RSTX**: convolutional layers from X onwards are unfrozen and reset to their starting values. **ERM-FT**: finetunes GDRO features using ERM. **SUBG-FT**: a group balanced subset of the data is used for finetuning. $L_{TR}$: Requires Group Labels during training; $L_{FT}$: Requires Group Labels during finetuning; **FR**: Retrains features; **FT**: Requires Finetuning. Finetuning a classifier on ERM features combining SUBG-FT and the GDRO loss retrieves 86% of GDRO's gains while not requiring group labels during training and using around 5% of the actual training data on the highest correlated version of the dataset.

| METHOD | 0.0 | 0.25 | 0.5 | 0.75 | 0.9 | $L_{TR}$ | $L_{FT}$ | FR | FT |
|---|---|---|---|---|---|---|---|---|---|
| ERM | 89.36% (0.55) | 89.72% (0.41) | 87.95% (0.63) | 81.62% (0.87) | 72.95% (0.36) | × | × | × | × |
| RW | 89.62% (0.24) | 90.91% (0.50) | 89.77% (0.63) | 88.06% (1.64) | 86.31% (0.30) | ✓ | × | × | × |
| GDRO | **90.86%** (0.18) | **91.69%** (0.09) | **90.45%** (0.09) | **89.30%** (0.09) | **87.31%** (0.27) | ✓ | × | × | × |
| GDRO-FT (FREE) | 90.65% (0.16) | 91.69% (0.36) | 90.24% (0.09) | 89.10% (0.82) | 83.80% (0.62) | × | ✓ | ✓ | ✓ |
| GDRO-FT (FRZ1) | 90.60% (0.09) | 91.64% (0.32) | 90.19% (0.16) | 88.89% (0.99) | 83.74% (0.85) | × | ✓ | ✓ | ✓ |
| GDRO-FT (FRZ2) | 90.65% (0.16) | 91.80% (0.09) | 90.24% (0.36) | 89.15% (0.63) | 83.90% (0.59) | × | ✓ | ✓ | ✓ |
| GDRO-FT (FRZ3) | 90.71% (0.18) | 91.80% (0.18) | 90.45% (0.32) | 89.25% (0.62) | 84.48% (0.80) | × | ✓ | ✓ | ✓ |
| GDRO-FT (FRZ4) | 90.60% (0.24) | **92.06%** (0.27) | **90.55%** (0.55) | 89.04% (0.24) | 84.90% (0.02) | × | ✓ | ✓ | ✓ |
| GDRO-FT (RST1) | **90.81%** (0.16) | 91.59% (0.16) | **90.45%** (0.09) | 89.24% (0.02) | **87.28%** (0.18) | × | ✓ | ✓ | ✓ |
| GDRO-FT (RST2) | 90.76% (0.24) | 91.59% (0.16) | 90.29% (0.24) | 89.01% (0.08) | 86.65% (0.08) | × | ✓ | ✓ | ✓ |
| GDRO-FT (RST3) | 90.45% (0.36) | 91.64% (0.32) | 89.98% (0.32) | 88.53% (0.09) | 85.49% (0.34) | × | ✓ | ✓ | ✓ |
| GDRO-FT (RST4) | 90.13% (0.39) | **92.06%** (0.27) | 90.03% (0.31) | 88.99% (0.18) | 84.56% (0.52) | × | ✓ | ✓ | ✓ |
| ERM-FT | 87.95% (0.48) | 87.51% (3.85) | 87.28% (2.03) | 87.73% (0.21) | 66.61% (2.96) | ✓ | × | × | ✓ |
| ERM-FT (FRZ1) | 88.32% (0.00) | 91.11% (0.54) | 87.33% (0.09) | 84.89% (0.41) | 73.10% (2.18) | ✓ | × | ✓ | ✓ |
| ERM-FT (FRZ2) | 88.11% (0.09) | 91.07% (0.24) | 87.28% (0.09) | 84.89% (0.41) | 73.16% (1.56) | ✓ | × | ✓ | ✓ |
| ERM-FT (FRZ3) | 88.47% (0.27) | 91.17% (0.18) | 87.69% (0.27) | 84.79% (0.59) | 73.00% (1.00) | ✓ | × | ✓ | ✓ |
| ERM-FT (FRZ4) | 88.58% (0.45) | 90.33% (0.96) | 88.16% (0.97) | 84.84% (0.32) | 71.60% (1.31) | ✓ | × | ✓ | ✓ |
| RW-FT | 84.32% (0.78) | 86.55% (0.50) | 82.71% (0.16) | 75.34% (1.15) | 66.20% (2.16) | × | ✓ | × | ✓ |
| GDRO-FT | 86.71% (0.50) | 87.28% (0.18) | 85.20% (0.56) | 79.08% (1.60) | 73.78% (2.92) | × | ✓ | × | ✓ |
| SUBG-FT | - | 90.76% (0.09) | 90.13% (0.55) | 89.56% (0.78) | 87.69% (1.09) | × | ✓ | × | ✓ |
| GDRO + GDRFR(GDRO) | - | **89.08%** (3.15) | **91.14%** (0.96) | 90.17% (0.42) | **83.33%** (1.33) | ✓ | ✓ | × | ✓ |
| GDRFR (RW) | - | 91.07% (0.24) | 91.12% (0.47) | **90.91%** (0.80) | 88.89% (0.78) | × | ✓ | × | ✓ |
| GDRFR (GDRO) | - | **91.43%** (0.41) | **91.48%** (0.59) | 90.86% (0.74) | **88.99%** (0.65) | × | ✓ | × | ✓ |

## B.1 UNFREEZING LAYERS WHEN FINETUNING USING GDRO ON WATERBIRDS

To understand the failure of just finetuning the classifier on Waterbirds, we finetune classifiers on ERM features using GDRO, but progressively unfreeze layers closer to the classifier. In this way, we progressively adjust the quality of the representation by way of GDRO's feature learning effects. Results for these experiments are shown in Table 2 under "GDRO-FT (FRZX/RSTX)", where X is a reference to which convolutional block onwards are being unfrozen. As we can see, by just unfreezing the final convolutional layer before the classifier we retrieve most of GDRO's performance. This suggests that most of GDRO's feature learning effects are taking place at the topmost layer. Better results come from restarting these layers instead of unfreezing them, with each restarted layer progressively adding a bit more performance, which suggests that ERM learned features are of lesser quality than GDRO features and that a small amount of impairment (Achille et al., 2019) has happened as we start from ERM features.

Table 3: Worst Group Accuracy results for various methods trained on all datasets (correlation=0.9) on different percentages of training data. All methods finetuned a classifier from frozen features derived from ERM training. Note that GDRFR already uses around 5-10% of the original training data. Even with 50% of their training data, very competitive results are obtained with GDRFR

| DATASET | METHOD | 0.1 | 0.25 | 0.5 | 0.75 | 0.9 | 1.0 |
|---|---|---|---|---|---|---|---|
| | ERM | - | - | - | - | - | 67.50% (1.18) |
| | RW-FT | 83.30% (0.75) | **83.87%** (0.81) | **84.60%** (0.66) | **84.73%** (0.35) | **85.40%** (0.30) | 82.61% (0.56) |
| MNIST-CIFAR | GDRO-FT | **83.50%** (0.85) | 83.77% (0.38) | 84.10% (0.44) | 84.63% (0.47) | 85.10% (0.20) | **82.74%** (0.99) |
| | GDRFR (RW) | 78.98% (5.45) | 82.73% (1.61) | 82.33% (0.70) | 82.61% (0.82) | 82.88% (1.28) | 82.46% (1.62) |
| | GDRFR (GDRO) | 78.19% (4.87) | 83.40% (0.83) | 82.95% (0.90) | 82.46% (1.41) | 82.72% (1.43) | **82.74%** (1.06) |
| | ERM | - | - | - | - | - | 72.95% (0.36) |
| | RW-FT | 49.43% (1.04) | 56.28% (2.16) | 63.29% (3.33) | 66.46% (2.21) | 66.25% (0.80) | 66.20% (2.16) |
| WATERBIRDS | GDRO-FT | 53.95% (2.74) | 62.41% (3.33) | 67.74% (0.68) | 73.26% (3.30) | 72.33% (1.15) | 73.78% (2.92) |
| | GDRFR (RW) | **81.15%** (7.18) | 85.15% (1.82) | 86.89% (0.45) | 88.36% (1.12) | 89.57% (0.19) | 88.89% (0.78) |
| | GDRFR (GDRO) | **81.15%** (7.18) | **85.49%** (2.22) | **87.27%** (0.72) | **88.72%** (1.15) | **89.75%** (0.05) | **88.99%** (0.65) |
| | ERM | - | - | - | - | - | 48.15% (2.10) |
| | RW-FT | 84.07% (0.85) | 84.63% (1.95) | 87.59% (0.32) | 88.52% (0.32) | 87.59% (1.28) | 88.61% (0.74) |
| CELEBA | GDRO-FT | **88.89%** (1.47) | **90.40%** (0.25) | **90.57%** (0.53) | **90.66%** (0.73) | **90.73%** (0.63) | **91.06%** (0.47) |
| | GDRFR (RW) | 84.74% (2.12) | 87.04% (1.16) | 87.00% (1.54) | 87.54% (0.69) | 88.03% (0.66) | 86.48% (1.40) |
| | GDRFR (GDRO) | 83.45% (3.81) | 87.59% (0.32) | 89.69% (0.33) | 89.73% (0.40) | 89.99% (0.52) | 90.10% (0.42) |

Table 4: DCI metrics for all datasets and methods studied.

| DATASET | CORR | DISENTANGLEMENT | | | COMPLETENESS | | | INFORMATIVENESS | | | | | |
|---|---|---|---|---|---|---|---|---|---|---|---|---|---|
| | | ERM | GDRO | RW | ERM | GDRO | RW | ERM | GDRO | RW | ERM | GDRO | RW |
| | 0.00 | 1.000 | 1.000 | 1.000 | 0.767 | 0.720 | 0.770 | 0.97 | 0.97 | 0.99 | 0.95 | 0.95 | 0.97 |
| | 0.25 | 0.997 | 0.998 | 0.998 | 0.737 | 0.719 | 0.739 | 0.97 | 0.97 | 0.99 | 0.95 | 0.95 | 0.96 |
| MNIST-CIFAR | 0.50 | 0.870 | 0.893 | 0.893 | 0.749 | 0.760 | 0.763 | 0.97 | 0.97 | 0.99 | 0.93 | 0.94 | 0.96 |
| | 0.75 | 0.406 | 0.411 | 0.294 | 0.809 | 0.719 | 0.836 | 0.98 | 0.98 | 0.99 | 0.88 | 0.91 | 0.92 |
| | 0.90 | 0.143 | 0.246 | 0.280 | 0.899 | 0.836 | 0.814 | 0.99 | 0.99 | 0.99 | 0.83 | 0.85 | 0.85 |
| | 0.00 | 0.981 | 0.972 | 0.957 | 0.545 | 0.528 | 0.484 | 0.97 | 0.96 | 0.96 | 0.92 | 0.92 | 0.89 |
| | 0.25 | 0.958 | 0.924 | 0.955 | 0.534 | 0.491 | 0.455 | 0.97 | 0.94 | 0.96 | 0.92 | 0.85 | 0.90 |
| WATERBIRDS | 0.50 | 0.902 | 0.891 | 0.857 | 0.529 | 0.513 | 0.443 | 0.97 | 0.97 | 0.96 | 0.88 | 0.89 | 0.85 |
| | 0.75 | 0.751 | 0.697 | 0.656 | 0.513 | 0.482 | 0.458 | 0.96 | 0.96 | 0.94 | 0.86 | 0.86 | 0.82 |
| | 0.90 | 0.400 | 0.296 | 0.460 | 0.478 | 0.407 | 0.430 | 0.96 | 0.94 | 0.95 | 0.81 | 0.78 | 0.76 |
| CELEBA | 0.90 | 0.961 | 0.940 | 0.918 | 0.561 | 0.541 | 0.538 | 0.97 | 0.96 | 0.96 | 0.94 | 0.93 | 0.93 |

Table 5: Percentage of gains from the best GRM over ERM for all methods that only finetune the classifier. We see that simple finetuning of the classifier retrieves most of the gains from the GRMs, except in the case of Waterbirds. GDRFR on the other hand, works very well on all datasets, in some cases, it even outperforms the GRMs performance. It also consistently outperforms SUBG-FT in every dataset and across levels of spurious correlations.

| DATASET | METHOD | CORRELATION | | | |
|---|---|---|---|---|---|
| | | **0.25** | **0.5** | **0.75** | **0.9** |
| MNIST-CIFAR | FT | 84,07% | **102,04%** | 99,20% | **94,38%** |
| | SUBG-FT | 94,10% | 85,04% | 100,42% | 93,91% |
| | GDRFR | **97,94%** | 92,12% | **100,84%** | **94,38%** |
| WATERBIRDS | FT | -123,86% | -110,00% | -33,07% | 5,78% |
| | SUBG-FT | 52,79% | 87,20% | 103,39% | 102,65% |
| | GDRFR | **86,80%** | **141,20%** | **120,96%** | **111,70%** |
| CELEBA | FT | - | - | - | **100,02%** |
| | SUBG-FT | - | - | - | 91,07% |
| | GDRFR | - | - | - | 97,79% |

# C   LIMITATIONS AND SOCIETAL IMPACT

Our analysis hinges on the following assumptions: groups within the dataset are usually unbalanced with the most represented groups benefiting from the spurious correlation. However, this is a standard setting in the robustness literature. Our theoretical analysis depends on a specific definition of a spurious vector, with which some may disagree.

Our proposed method requires extra data with annotations of both the class and spurious label for the finetuning stage. It does not require much of it, 5% for the full method, but Table 3 shows decent results with even less. All competing baselines require at least this much extra data.

We believe this work may have a positive societal impact as it is trying to mitigate a problem of spurious correlations that usually happens on underrepresented subsets of the data, which may represent in practice underrepresented parts of society. Moreover, our work seeks to understand the mechanisms by which GRMs work which could aid in developing better methods that work to achieve robustness and fairness.

## D    EXPERIMENT DETAILS

### D.1    HYPERPARAMETERS

#### D.1.1    GENERAL

All methods use SGD with momentum 0.9 as their optimizer. We use L2 regularization and no data augmentation. No learning rate scheduler is used. All methods are run for 3 seeds. Learning rates and weight decay used were taken from (Liu et al., 2021), which were based off of (Sagawa* et al., 2020).

#### D.1.2    MNIST-CIFAR

We train all models for 5000 epochs/iterations. L2 regularization of $10^{-4}$, learning rate of 0.001 for all methods. Batch size is 10000.

#### D.1.3    WATERBIRDS

We train all models for 300 epochs. Batch size is 64. For ERM, we use L2 regularization of $10^{-4}$, learning rate of $10^{-4}$; for GDRO L2 regularization of 1, learning rate of $10^{-5}$; for RW, L2 regularization of $10^{-3}$, learning rate of $10^{-4}$.

#### D.1.4    CELEBA

We train all models for 50 epochs. Batch size is 64. For ERM, we use L2 regularization of $10^{-4}$, learning rate of $10^{-4}$; for GDRO L2 regularization of 0.1, learning rate of $10^{-5}$; for RW, L2 regularization of 0.1, learning rate of $10^{-5}$.

## E    CO2 EMISSION RELATED TO EXPERIMENTS

Experiments were conducted using a private infrastructure, which has a carbon efficiency of 0.432 kgCO$_2$eq/kWh. A cumulative of 2275 hours of computation was performed on hardware of type GTX 1080 Ti (TDP of 250W).

Total emissions are estimated to be 245.7 kgCO$_2$eq of which 0 percents were directly offset.

Estimations were conducted using the MachineLearning Impact calculator presented in Lacoste et al. (2019).

## F    DERIVATIONS AND PROOFS

### F.1    DERIVATION OF $\frac{\partial \mathcal{L}}{\partial W}$ FOR EACH METHOD

#### F.1.1    ERM

In standard ERM, our loss function is simply: $\sum\limits_{i=1}^{n} \frac{\mathcal{L}(x_i, y_i)}{N}$

$$\frac{\partial \mathcal{L}}{\partial W} = \frac{\partial \mathcal{L}}{\partial z} \cdot \frac{\partial z}{\partial W}$$

Where $s_i$ is the softmax output of $x_i$, $\frac{\partial \mathcal{L}}{\partial z} = s_i - y_i$, $\frac{\partial z}{\partial W} = x_i$ and $z_i = W \cdot x_i + b$.

Therefore:

$$\frac{\partial \mathcal{L}(W, x_i, y_i)}{\partial W} = (s_i - y_i) \cdot x_i$$

When finetuning our classifier with static features:

$$\Delta W = \sum_{i=1}^{N} \frac{(s_i - y_i) \cdot x_i}{N}$$

That is $\Delta W$ is just a weighted average of the representations with more weight given to samples with greater error.

### F.1.2 REWEIGHTING

In Reweighting, our loss function becomes:

$$\sum_{g \in \mathcal{G}} \sum_{x_i \in \mathbb{X}_g} \frac{\mathcal{L}(x_i, y_i)}{N_g}$$

That is to say, it becomes the sum of the average loss within each group. Now, our update becomes:

$$\Delta W = \sum_{g \in \mathcal{G}} \sum_{x_i \in \mathbb{X}_g} \frac{(s_i - y_i) \cdot x_i}{N_g}$$

$$\Delta W = \sum_{g \in \mathcal{G}} \bar{x}_g$$

Let's call $\sum_{x_i \in \mathbb{X}_g} \frac{(s_i - y_i) \cdot x_i}{N_g} = \bar{x}_i$ the weighted average representation of group $i$, where representations gain more weight the more the classifier is wrong. $\Delta W$ is just a weighted average of the $\bar{x}_i$!

### F.1.3 GROUP DRO

In Group DRO, our loss function becomes:

$$\sum_{g \in \mathcal{G}} p_g \cdot \mathcal{L}_g$$

With:

$$p_g = \frac{e^{\epsilon \cdot \mathcal{L}_g}}{\sum\limits_{i=1}^{G} e^{\epsilon \cdot \mathcal{L}_i}}$$

Since $p_g$ depends on $W$, using the product rule of derivatives we get:

$$\frac{\partial \mathcal{L}}{\partial W} = \sum_{g \in \mathcal{G}} \left( \frac{\partial p_g}{\partial W} \cdot \mathcal{L}_g + p_g \cdot \frac{\partial \mathcal{L}_g}{\partial W} \right)$$

The second term on the right is equivalent to the Reweighting derivative we calculated previously. Let $z_g = \epsilon \cdot \mathcal{L}_g$ and $Z = [z_1, ..., z_G]$

$$\frac{\partial p_g}{\partial W} = \frac{\partial p_g}{\partial Z} \cdot \frac{\partial Z}{\partial W}$$

From the derivative of Softmax:

$$\frac{\partial p_g}{\partial z_g} = p_g \cdot (\mathbb{1}_{g=j} - p_j)$$

$$\frac{\partial z_g}{\partial W} = \epsilon \frac{\partial \mathcal{L}_g}{\partial W}$$

$$\frac{\partial \mathcal{L}_g}{\partial W} = \sum_{x_j \in \mathbb{X}_g} \frac{(s_j - y_j) \cdot x_j}{N_g} = \bar{x}_g$$

Then:

$$\frac{\partial p_g}{\partial W} = \frac{\partial p_g}{\partial Z} \cdot \frac{\partial Z}{\partial W} = \epsilon \sum_{j \in \mathcal{G}} p_g \cdot (\mathbb{1}_{g=j} - p_j) \cdot \bar{x}_j$$

Finally:

$$\frac{\partial \mathcal{L}}{\partial W} = \epsilon \sum_{g \in \mathcal{G}} \sum_{j \in \mathcal{G}} p_g \cdot (\mathbb{1}_{i=j} - p_j) \cdot \bar{x}_j \mathcal{L}_g + \sum_{g \in \mathcal{G}} p_g \cdot \bar{x}_i$$

$$\frac{\partial \mathcal{L}}{\partial W} = \epsilon \sum_{g \in \mathcal{G}} \sum_{j \in \mathcal{G}} p_j \cdot (\mathbb{1}_{g=j} - p_g) \cdot \bar{x}_g \mathcal{L}_j + \sum_{g \in \mathcal{G}} p_g \cdot \bar{x}_g$$

$$\frac{\partial \mathcal{L}}{\partial W} = \epsilon \sum_{g \in \mathcal{G}} \bar{x}_g \sum_{j \in \mathcal{G}}^{G} p_j \cdot (\mathbb{1}_{g=j} - p_g) \mathcal{L}_j + \sum_{g \in \mathcal{G}} p_g \cdot \bar{x}_g$$

Let's define $l_g = \epsilon \sum_{j \in \mathcal{G}} p_j \cdot (\mathbb{1}_{i=j} - p_g) \mathcal{L}_j$:

$$\frac{\partial \mathcal{L}}{\partial W} = \sum_{g \in \mathcal{G}} \bar{x}_g \cdot l_g + \sum_{g \in \mathcal{G}} p_g \cdot \bar{x}_g$$

$$\frac{\partial \mathcal{L}}{\partial W} = \sum_{g \in \mathcal{G}} \bar{x}_g \cdot (l_g + p_g)$$

It's also noteworthy to understand when the $l_g > 0$:

$$l_g = \epsilon \sum_{j \in \mathcal{G}} p_j \cdot (\mathbb{1}_{g=j} - p_g) \mathcal{L}_j > 0$$

Dividing by $p_g$ and rearranging terms, we get the following inequality:

$$(1 - p_g) \cdot \mathcal{L}_g > \sum_{g \neq j} p_j \cdot \mathcal{L}_j$$

$$(\sum_{g \neq j} p_j) \cdot \mathcal{L}_g > \sum_{g \neq j} p_j \cdot \mathcal{L}_j$$

This condition states in conventional language that if the expected loss if we made the $g-$th group as important as the other groups is greater than the actual expected loss of the rest of the groups. So basically, this condition is forcing the average loss to be the same for all groups.

### F.2 PROOF FOR THEOREM 4.1

Recall the definitions of Theorem 4.1:

$$\frac{\Delta \alpha_{sp}}{\Delta t}^{ERM} = \sum_{g \in \mathcal{G}} \sum_{j=1}^{|\mathbb{X}_g|} \frac{\lambda_j \cdot (s_j - y_j)}{N}$$

$$\frac{\Delta \alpha_{sp}}{\Delta t}^{RW} = \sum_{g \in \mathcal{G}} \sum_{j=1}^{|\mathbb{X}_g|} \frac{\lambda_j \cdot (s_j - y_j)}{4 N_g},$$

where $\lambda_j = \vec{v}_{sp} \cdot x_j$, $N_g$ is the size of group $g$ and $N$ is the total amount of samples. Then, $\frac{\Delta \alpha_{sp}}{\Delta t}^{ERM} > \frac{\Delta \alpha_{sp}}{\Delta t}^{RW}$.

*Proof.* Recall that our dataset is divided in 4 groups $\mathcal{G} = \{s0, s1, n0, n1\}$, which are groups defined by whether the spurious correlation is aligned with the ground truth label or not.

For the standard spurious correlation setting we have:

$$\left( \frac{1}{N} - \frac{1}{4 \cdot N_{sj}} \right) > 0$$
$$\left( \frac{1}{N} - \frac{1}{4 \cdot N_{nj}} \right) < 0$$

where $j \in \{0, 1\}$.

Now, let $\vec{v}_{sp}$ be a spurious vector with respect to $G$ as defined in Definition 2.1 , then for the $j$-th example we have $\lambda_j = \vec{v}_{sp} \cdot x_j$.

If we denote:

$$\bar{x}_g = \sum_{i=1}^{|\mathbb{X}_g|} \frac{(s_i - y_i) \cdot x_i}{N_g},$$

then $\dfrac{\Delta \alpha_{sp}}{\Delta t}^{RW} = \vec{v}_{sp} \cdot \left( \sum_{g \in \mathcal{G}} \bar{x}_g \right)$ . Let us define for all $g \in \mathcal{G}$:

$$w_g^{ERM} = \frac{1}{N} \sum_{j=1}^{|\mathbb{X}_g|} \lambda_j \cdot (s_j - y_j)$$

$$w_g^{RW} = \frac{1}{4 \cdot N_g} \sum_{j=1}^{|\mathbb{X}_g|} \lambda_j \cdot (s_j - y_j)$$

So,

$$\frac{\Delta \alpha_{sp}}{\Delta t}^{ERM} = \sum_{g \in \mathcal{G}} w_g^{ERM}$$

$$\frac{\Delta \alpha_{sp}}{\Delta t}^{RW} = \sum_{g \in \mathcal{G}} w_g^{RW}$$

Let $w_g^{\Delta} = w_g^{ERM} - w_g^{RW}$. Replacing and reordering we conclude that the condition we want to prove becomes:

$$\sum_{g \in \mathcal{G}} w_g^{\Delta} > 0$$

Finally, note that:

$$w_g^{\Delta} = \sum_{j=1}^{|\mathbb{X}_g|} \lambda_j \cdot (s_j - y_j)(\frac{1}{N} - \frac{1}{4 \cdot N_g})$$

is positive for all $g$ by the definition of $\lambda_j$ and the spurious correlation setting assumptions. Thus, $\frac{\Delta \alpha_{sp}}{\Delta t}^{ERM} > \frac{\Delta \alpha_{sp}}{\Delta t}^{RW}$ . $\qquad\square$

