# OpenReview forum: "Towards Understanding Why Group Robustness Methods Work"
_ICLR.cc/2025/Conference — Submitted to ICLR 2025_

### Official Review · Reviewer_XQ6v · 2024-10-31

**Soundness:** 3
**Presentation:** 2
**Contribution:** 2
**Rating:** 3
**Confidence:** 4

**Summary:**

This paper proposes to study the underlying properties of a class of "group robustness" methods, including reweighting and group distributionally robust optimization (GDRO). They explore several hypotheses around the embeddings and classifier layers learned by these approaches as compared to ERM, finding evidence that GDRO representations both include group information and that it is more entangled in the embeddings layer. They propose a method for group-robust classification using group labels for finetuning leveraging these GDRO-related insights.

**Strengths:**

- some interesting insights: in particular, explorations around disentanglement I believe are a novel set of questions and effects
- systematic exploration of various methods in Table 1 is worthwhile
- Fig 2, particularly in a), is a nice demonstration of the difference in classifier behavior
- the result in 4.1 around the cosine similarity gradient being larger for ERM is surprising and interesting, I think the authors could delve into this (and the GDRO note) a little more deeply

**Weaknesses:**

- I believe one of the central results in this paper is not novel - the observations around GDRO embeddings containing sensitive information, and the efficacy of finetuning methods on top of ERM embeddings, have been shown before. The first paper I am aware of doing this is “Overparameterisation and worst-case generalisation: friend or foe?”, Menon et al (ICLR 2021)
- The definition of “spurious vector” is unclear to me - it seems to suggest that for all x in s0/n1, we have lambda > 0. This seems overly strong: might there not be some outlier x’s for which this doesn’t hold? A definition centered on an expectation seems more reasonable to me
- I’m a little confused by the definition of GDRO on line 190 - this doesn’t seem to match the definition from the Sagawa paper, which uses a min/max formulation over groups. Is there some other definition I should be looking at?
- Fig 1/Sec 3 shows a good result, but I’m confused why we need to look at singular vectors - wouldn’t prediction for the spurious label -from the embedding itself be a more useful number?
- The proposed method GDRFR is never really explained, just the sentence “Therefore, finetuning a classifier on features derived from ERM combining SUBG-FT and GDRO” is given. This seems like a big weakness in the paper - I don’t know what the proposed method of this section does! There should be a clear explanation of this approach
- I think the entanglement results are quite interesting, however in Fig 3 the effects don’t actually appear to be very significant - the trends don’t seem either so large or consistent. As a major part of the paper, I would love to see either a) a clearer empirical distinction, or b) an argument around why these results are actually significant (either statistically, or explain why a difference of this magnitude in the places we see it actually does matter)

Smaller points:
- I strongly dislike the s0, s1, n0, n1 notation - I prefer “01” (e.g. value of s and value of y) notation. The s/n notation is confusing to me - e.g. I believe on L123 the definition is incorrect due to this confusion (should be s0 and s1 where the vector is spurious, since those are where the correlation holds)
- It’s not clear to me what the label is in the MNIST-CIFAR task - can use some more careful description
- Line 268: should this gradient be wrt W^t? I would have thought that t is discrete
- Line 282 - a little unclear on notation - I thought p was indexed by “i” but here it’s indexed by g (which seems different)
- L322: is the inequality in the wrong direction here (typo)?
- The choice to put the CelebA data in the 0.9 column in Table 1 is odd - at the very least I think you should put a note in the caption about what’s going on there
- I think SUBG is never defined - not sure what that is

**Questions:**

Summarizing Suggestions:
- a few unclear definitions that need cleaning up (GDRFR, GDRO, "spurious vector")
- more discussion on significance of entanglement results

General Extra Questions:
- It would be helpful to have a little more intuition on how to think about \ell_g (line 285) - seems like an interesting quantity but hard to interpret
- I’m curious why the authors think the results in 2b are so sharply rising at 0.9
- “ERM has no incentive to use the invariant features” (L365) - I see what this is gesturing at but could use some fleshing out, I think it’s an important and interesting point!

---

### Official Review · Reviewer_uCsT · 2024-11-04

**Soundness:** 3
**Presentation:** 2
**Contribution:** 3
**Rating:** 6
**Confidence:** 3

**Summary:**

The paper investigates the mechanisms that enable GRMs to achieve robustness against spurious correlations in deep learning models. The paper suggests that GRMs achieve robustness by learning a linear classifier that ignores spurious feature. Moreover, they learn representations distributing prediction across multiple features rather than relying on spurious ones. An improvement called Group Distributionally Robust Feature Reweighting (GDRFR) is proposed, which retains most of the benefits of GRMs with limited use of group labels in the finetuning phase.

**Strengths:**

The mechanism of GRMs is still unknown so the paper investigates an important and interesting problem. With both empirical and theoretical analysis, the result of the paper is solid. The logical chain of this paper is very clear. The authors systematically analyzed several possible mechanisms by comparing ERM and GRMs. By rejecting some hypothesis and verifying others, the paper explains the success of GRMs in spurious correlation setting.

**Weaknesses:**

The presentation of this paper is not satisfactory to me. Here are several concerns about this paper:
1. Some of the definitions are not very clear. For example the four groups $\{s0,s1,n0,n1\}$, it is quite abstract if only state the definition without an example. I am not quite sure what are these groups. Take waterbird dataset as an example, in my understanding $s0$ means waterbird + water background, $s1$ means landbird + land background, $n0$ means waterbird + land background and $n1$ means landbird + water background. Is this correct? I think with some examples like this, it would be more clear.
2. The statement in Theorem 4.1 is not standard. In your Thm 4.1, it seems like you define $\frac{\Delta\alpha^{ERM}}{\Delta t}$ and $\frac{\Delta\alpha^{RW}}{\Delta t}$ by the formulas in line 291-296 and with these definitions you got the inequality between them. This is a useless statement if we define these two terms like that. However, I understand that the actual definitions of these terms are located in line 268. I think the formulas in line 291-296 should be stated as a result derived from line 268 together with the computations in line 274-282, instead of a definition, making this statement meaningful.
3. This paper proposed a method called GDRFR, but I didn't see the explanation of the method. I think at least a short introduction should be included in the main paper.

**Questions:**

See weakness part.

---

### Official Review · Reviewer_12wh · 2024-11-05

**Soundness:** 1
**Presentation:** 1
**Contribution:** 1
**Rating:** 3
**Confidence:** 4

**Summary:**

The authors make three observations regarding how group robustness methods (GRM) work as opposed to empirical risk minimization (ERM) methods: 1- Both methods learn the spurious features and retain them their learned representations, 2- GRM work through creating a linear classifier that ignores spurious information 3- ERM creates more disentangled representations, while GRM leads to models that rely on multiple features. Based on these insights the authors propose a new baseline method for group robustness research.

**Strengths:**

- Understanding why methods that achieve robustness against spurious correlations at the representation level is a crucial endeavour for developing robust machine learning methods.
- The authors' observation about how ERM and GRM representations differ in terms of disentanglement-related metrics could be helpful for further understanding the mechanisms through which robustness against spurious correlations is obtained.

**Weaknesses:**

- I cannot see the novelty of the authors' first set of results. The paper by Izmailov et al. 2022 already states that "These results suggest that the improvements over ERM in base model performance for methods such as group DRO and RWG are largely the result of better weighting of the learned features rather than learning better representations of the core features." Is it surprising to see that the spurious features are retained in the last layer? The authors do not elaborate enough on these results to make this observation a marginally significant contribution, or discuss them in relation to novel findings (e.g. those that show features can be learned even if they are completely uninformative of the true label; Bombari and Mondelli, 2024).
- The criticism above applies to their second contribution. Izmailov et al. (2022) already show that GRMs achieve improved results by leading to linear classifiers that are more sensitive to core feature representations vs. spurious feature representations. What is the significance of the observation that this is achieved by weights being orthogonal to spurious feature representations? Does this have any important downstream consequences than if the mechanism was slightly different?
- The paper is hard to follow due to notational and terminological difficulties and unjustified decisions. I detail these observations in the next section.


Bombari, S., & Mondelli, M. (2024). How Spurious Features are Memorized: Precise Analysis for Random and NTK Features. Proceedings of the 41st International Conference on Machine Learning, 4267–4299. https://proceedings.mlr.press/v235/bombari24a.html

**Questions:**

- L013: Incomplete sentence.
- L047: "optimize to maximize"
- L096: What does "retrieving GRMs feature learning capabiliites" mean? Please be clearer.
- L113: "whether the spurious correlation is predictive of the ground truth label". Did you mean spurious feature(s)?
- L122: What does $\vec{x} \in \{s0, n1\}$ mean? I think this notation is problematic.
- L124: How is $\vec{v}$ not useful in groups where the spurious correlation is nonexistent? Would not a binary classifier that outputs $1$ if $\lambda > 0 $ and $0$ otherwise classify correctly in groups $n0, n1$?
- L132: How's correlation defined here?
- L150: What is the justification for the selected models?
- L190: What is $\mathcal{L}_i$?
- L192: $\mathbb{X}_g$ is undefined? What is the source of $y_i$? In L263 $\mathbb{X}_g$ is defined as the _image_ of x belonging to a certain group, which seems to not fit its usage here, making this more confusing.
L194: What does FT in SUBG-FT refer to here? Fine-tuning? This is not discussed when describing the method.

---

### Official Review · Reviewer_9nHj · 2024-11-07

**Soundness:** 2
**Presentation:** 3
**Contribution:** 2
**Rating:** 3
**Confidence:** 4

**Summary:**

The paper studies mechanisms by which group DRO and reweighting methods discard spurious correlations when training deep networks with gradient based methods.

In terms of formal results, it is shown that for each iteration of gradient descent the change in the projections of the last layer weights on directions that encode spurious features, is smaller for methods that reweigh groups than it is for ERM (and the this holds for GroupDRO and ERM under some conditions). Empirically, it is shown that the last layer of deep networks trained with group robustness methods still encodes spurious features but that the representations learned by group-robustness methods are less axis-aligned than those learned by ERM. It is claimed that these more distributed representations are helpful towards robustness.

**Strengths:**

The paper touches upon an interesting problem, which is how do different group-robustness methods affect the learned classifier and representations. The results are intuitive and experiments are executed well.

**Weaknesses:**

I think that while the paper has some nice results, it does not contribute substantial novel insights. The formal results mainly touch upon linear models, where a lot is known about group robustness methods, while they overlook some important aspects of the problem.

For instance, the theorem about gradient directions is intuitive and seems like a straightforward consequence of reweighting a loss. It doesn't tell us much about the dynamics of representation learning, which the authors only explore empirically.
The result is also rather limited, it just says that the weight placed by the robust methods on the spurious features will be smaller than that ERM puts on them, it doesn't say whether the weight will be small with respect to the weight placed on the robust feature (which is what leads to high accuracy and robustness in unison).
This is important, one reason is that it is well known that robust methods fail once they are trained with large models without controlling for overfitting, e.g. [1, 2]. Hence, having a smaller weight on spurious features than ERM is not a sufficient result, and if we train SGD for enough iterations the dependence on the spurious feature will grow large. Usually in benchmarks such as those the authors experiment with, the problem is addressed by including large $\ell_2$ regularization or early stopping, or alternatively with 2-stage methods such as deep feature reweighting that avoid overfitting by fixing the feature extractor. These are essential to the success of group robustness methods, but the theory does not analyze them.

The authors also mention that the last layer encoding the spurious feature is a surprising property. I did not understand why this is a surprise, and even if it is surprising, I'd be much more interested in understanding what in the training dynamics causes that (e.g. by an analysis of a 2-layer network). Finally, the comment about "completeness" being higher for ERM than the robust methods might be interesting, but I did not understand whether the difference of about 3-5% is that meaningful, and also why is it smaller in MNIST-CIFAR for spurious correlations that aren't very strong.

[1] Wald, Y., Yona, G., Shalit, U., & Carmon, Y. Malign Overfitting: Interpolation can Provably Preclude Invariance. In The Eleventh International Conference on Learning Representations.‏

[2] Zhai, R., Dan, C., Kolter, J. Z., & Ravikumar, P. K. Understanding Why Generalized Reweighting Does Not Improve Over ERM. In The Eleventh International Conference on Learning Representations.‏

**Questions:**

For results in figure 1, is there a reason for taking the svd of representations and then predicting from each vector, instead of simply predicting the spurious feature from the representation?

The notation in the equation at line 268 wasn't entirely clear to me. The right hand side has a multiplication of a vector $\mathbf{v}_{sp}$, a scalar $\mu$, and another vector $\frac{\partial \mathcal{L}}{\partial W^t}$. It'd be helpful to emphasize which multiplications are scalar multiplication, which ones are inner products etc.

In the first case in remark 4.1 it is stated that the condition "will usually hold as the group will be overrepresented in the dataset, which should make its average loss be below the total average loss". This is only true if it is as easy to achieve low loss on the overrepresented group as it is on the other groups. It might be useful to make such a general claim more precise.

---

### Official Review · Reviewer_WveP · 2024-11-10

**Soundness:** 2
**Presentation:** 2
**Contribution:** 1
**Rating:** 3
**Confidence:** 4

**Summary:**

This paper aims to understand the reasons for the effectiveness of group reweighting methods on subgroup robustness. It analyzes the representations and classifiers of neural nets trained with group DRO and reweighting methods compared to ERM, on image datasets. Based on the observation that ERM learns more disentangled representations while GDRO learns a classifier more orthogonal to the spurious feature directions, they propose a new method called GDRFR that is more efficient and has a similar performance as group robustness methods.

**Strengths:**

The paper formalizes and quantifies some of the findings of [1]. This paper uses the framework of DCI metrics (disentanglement, completeness, and informativeness) to show that representations learned by ERM are more disentangled compared to GDRO and reweighting.

**Weaknesses:**

The paper has some major weaknesses.

1. While understanding how group robustness methods work is an important research area, the findings in this paper don’t provide new insights in this direction, as they are quite similar to [1]. [1] showed that the representations learned by ERM are superior to other group robustness methods since re-training only the last layer on a balanced set of labeled data can lead to good subgroup robustness. Hence, it is not surprising that ERM learns better representations while group robustness methods learn better classifiers.
2. The evaluation is done in an extremely limited setting. The authors only consider two group robustness methods, namely reweighting and GDRO, and only consider three image datasets, namely CelebA, Waterbirds, and MNIST-CIFAR. This is a concern for two reasons:
    1. The paper is closely related to [1] (as mentioned above), and [1] considers a broader setting with more datasets and methods.
    2. Some of the findings are inconsistent on the datasets. Since the main contribution of the paper seems to be quantifying the findings in [1], it should do a more thorough evaluation.
3. The proposed method only gets gains on the Waterbirds dataset. It should be compared on more datasets and with more methods, e.g., those considered in [1].
4. There are several missing references in the discussion on related work, e.g. [2-5].
5. (Minor) Issues with the writing:
    1. There are several instances where technical terms, e.g., "Empirical Risk Minimization,"  are capitalized in the middle of a sentence. The authors should review the paper to ensure uniform formatting for terms for clarity and consistency.
    2. The equations use up more space than required. The spacing can be reduced.

References:

[1] P. Kirichenko, P. Izmailov, and A. Wilson. Last layer re-training is sufficient for robustness to spurious correlations. In ICLR 2023.

[2] Y. Deng, Y. Yang, B. Mirzasoleiman, and Q. Gu. Robust learning with progressive data expansion against spurious correlation. In NeurIPS 2023.

[3] B. Vasudeva, K. Shahabi, and V. Sharan. Mitigating simplicity bias in deep learning for improved OOD generalization and robustness. In TMLR 2024.

[4] J. Nam, H. Cha, S. Ahn, J. Lee, and J. Shin. Learning from failure: Training debiased classifier from biased classifier. In NeurIPS 2020.

[5] A. Setlur et al. Bitrate-constrained DRO: Beyond worst-case robustness to unknown group shift. 2023.

**Questions:**

Please see the weaknesses section, especially the first three points.

---

### Meta-Review · Area_Chair_k1RC · 2024-12-20

**Metareview:**

The paper explores the mechanisms behind Group Robustness Methods (GRMs) and their ability to mitigate spurious correlations compared to Empirical Risk Minimization (ERM). The questions of understanding how to get better group robustness is interesting and important, but the findings are not differentiated substantially from prior work, such as Kirichenko et al. 2023. The main concern is novelty, but there were also some concerns regarding the empirical evaluation and writing. Therefore, I do not believe the paper meets the bar in its present form.

**Additional Comments On Reviewer Discussion:**

There was no rebuttal from authors.

---

### Decision · Program_Chairs · 2025-01-22

Reject